# Quasi-Static Rheological Properties of Lithium-Based Magnetorheological Grease under Large Deformation

**DOI:** 10.3390/ma12152431

**Published:** 2019-07-30

**Authors:** Huixing Wang, Guang Zhang, Jiong Wang

**Affiliations:** 1School of Mechanical Engineering, Nanjing University of Science and Technology, Nanjing 210094, China; 2School of Civil and Environmental Engineering, University of Technology Sydney, Ultimo 2007, Australia

**Keywords:** MR grease, quasi-static monotonic/cyclic shear, larger deformation, stress, damping

## Abstract

This paper investigates the quasi-static rheological properties of lithium-based magnetorheological (MR) grease under large deformation. Three types of lithium-based MR grease comprising different mass ratios of carbonyl iron (CI) particles and lithium-based grease were prepared. The dependence of the magneto-induced stress–strain curves for MR grease on CI particles content, shear rate, and shear deformation under quasi-static monotonic shear conditions were tested and discussed. The results demonstrate that the shear rate dependence of the maximum yield stress is significantly weakened by the magnetic field, and this weakening is further enhanced as the CI particles content of MR grease increases. In addition, to evaluate and characterize the behavior of the cyclic shear–stress curves of MR grease under quasi-static condition, cyclic shear tests under different controlled conditions, i.e., CI particles content, shear rate, shear strain amplitude, and magnetic field strength, were conduct and analyzed. The magneto-induced shear stress of MR grease with higher CI particles content shows a sharp decrease during the transition from loading to unloading. Moreover, the experiment results also show that the damping characteristics of MR grease are highly correlated with CI particles content, shear strain, and magnetic field strength.

## 1. Introduction

Magnetorheological (MR) grease is a new type of magneto-induced smart material composed of soft magnetic particles and grease matrix, which exhibits a transition state between traditional MR fluid (liquid state) [1,2,3] and MR elastomer (solid-state) [4,5,6] in the absence of a magnetic field. Compared with the severe problems of sedimentation and leakage faced by liquid MR fluid [2,7,8], the MR grease has advantages of high stability and low leakage [9]. Moreover, under the stimulation of external magnetic field, the magnetic particles in MR grease can break through the restraint of the grease matrix and form an ordered chain or cluster structure, which is expected to possess a higher magneto-induced effect than MR elastomer [9,10]. In a word, the excellent features of MR grease could upgrade the performances of the MR device, i.e., MR damper [11,12], MR clutch [13], MR isolator, and MR actuator [14,15], compared with the use of MR fluid.

For MR grease, most researches focus on the rheological properties, such as viscosity, yield stress, storage modulus, loss modulus, and loss factor under different magnetic field strength and temperature. Rankin et al. investigated the yield stress of MR grease with 10 vol% of CI particle suspensions. They found that the yield stress of MR grease increased sub-quadratically with the external magnetic flux density [16]. Sahin et al. tested the rheological properties of MR grease under steady shear test. The experimental results showed that the yield stress of MR grease is higher than that of traditional MR fluid, but the off-state viscosity is higher [17]. Gordaninejad et al. studied the temperature characteristics of MR grease under various magnetic field strengths; the results demonstrated that the temperature dependence of MR grease is more pronounced than MR fluid [18]. Choi et al. prepared MR grease by dispersing soft magnetic particles in a grease medium and examined its MR characteristics. The results proved that MR grease exhibit a strong solid-like structure with the applied magnetic field [19]. Mohamad et al. studied the magnetic-induced properties of MR grease with varies weight percentages of the magnetic particles. The results demonstrated that the viscosity and storage modulus rapidly increase with the external magnetic field, and the maximum yield stress and relative MR effect are as high as 52.7 kPa and 952.38% respectively [20]. Wang et al. tested the field-dependent characteristics of MR grease under different temperatures. They found that the temperature dependence of rheological properties of MR grease decrease with the increase of the external magnetic field [10]. Most studies on the characteristics of MR grease are limited to dynamic shear test. The rheological properties demonstrated by MR grease under quasi-static shear test have been rarely investigated.

As known, in the practical application of MR devices, the external load applied on MR devices can be divided into three types: static load, quasi-static load, and dynamic load, among which dynamic load is the most common. However, in the case of vibration reduction of bridges and buildings, MR dampers often work under periodic quasi-static loads [21,22,23,24]. At this time, MR materials inside the damper are subjected to large deformation and quasi-static shearing. The field-induced properties of MR materials under quasi-static shear will determine the performance of MR damper used in bridges and buildings. Therefore, it is especially necessary to carry out investigations on quasi-static rheological properties of MR grease under large deformation to promote engineering application.

In this paper, the rheological properties of MR grease with different weight percentages of carbonyl iron (CI) particles under quasi-static shear mode and large shear deformation were investigated. Firstly, MR grease with 30%, 50%, 70% mass fraction of CI particles was prepared, respectively. The filed-induced stress of MR grease under the quasi-static monotonic shear condition with different shear rate and CI particles content were tested and discussed. Finally, the dependence of the stress–strain curves and damping performance for MR grease on shear rate, magnetic field strength, CI particles content under quasi-static cyclic shear condition were calculated and analyzed.

## 2. Experimental Testing

### 2.1. MR Grease Preparation

In this paper, MR grease samples with various weight percentages of CI particles were prepared by employing lithium-based grease as a continuous phase. The fabrication process of the MR grease includes three steps: First, a specified amount of CI particles (type CN, BASF, Ludwigshafen, Germany) with average diameter of 6 μm and lithium-based grease (Gadus S2 V220, Shell (China) Ltd., Beijing, China) with NLGI 0 are weighed separately and placed into two different vials. Second, lithium-based grease was agitated with a mechanical agitator at 500 rpm for 10 min after the temperature reached 80 °C. Finally, the CI particles were added into the vial and mixed with the grease matrix at the rotating speed of 800 rpm until thoroughly mixed.

Three types of MR grease with different CI particles were prepared in this experiment and are shown in Table 1. According to the mass fraction of CI particles in each MR grease, the samples were named as MRG-30, MRG-50, and MRG-70, respectively, i.e., MRG-70 represents 70 g of CI particles per 100 g MR grease. Figure 1 is the sample of MRG-70, which exhibits a semi-solid state in the zero-field condition.

### 2.2. Quasi-Static Test of MR Grease

The quasi-static properties of MR grease were measured by a parallel-plate rheometer (type MCR 302, Anton Paar Co., Graz, Austria) equipped with the magnetorheological module (Type MRD180, Anton Paar Co., Austria) and temperature-control accessory (Type F25, Julabo Technology Co., Seelbach, Germany) through the quasi-static monotonic shear and cyclic shear test. The sketch map of MCR 302 rheometer is shown in Figure 2. In this experiment, MR grease was placed in the gap between the base mount and parallel-plate measuring tool. The parallel-plate had a diameter of 20 mm, and its distance from the base was always set to 1 mm. The magnetic field strength passing through the gap increased from 0 kA/m to 866 kA/m when regulating the coil current change from 0 A to 5 A.

In quasi-static monotonic shear tests, the linear strain signal shown in Figure 3a was applied to the sample through the rheometer using displacement control. The maximum shear strain amplitude in this test maintained at 100%. The time used for increasing shear strain linearly from 0 to 100% was set to 10 s, 20 s, 30 s, 50 s, and 200 s, respectively. And thus, the corresponding quasi-static shear rates were 0.1/s, 0.05/s,0.033/s, 0.02/s and 0.005/s, respectively.

The dependence of the cyclic stress–strain curves for MR grease on CI particles content, shear rate, shear strain amplitude under different magnetic field strength were tested by using a repeated triangular wave, as shown in Figure 3b. In the cyclic shear test for MR grease with different CI particles content, three types of MR grease, i.e., MRG-30, MRG-50, and MRG-70, were used as the samples and tested at a fixed shear strain, 100%, and shear rate, 0.05/s. In the cyclic shear test for MRG-70 under different shear rate, the shear strain was fixed at 100%, and the shear rate was set as 0.005/s, 0.02/s, 0.033/s, and 0.1/s, respectively. In the cyclic shear test for MRG-70 under different shear strain, the shear strain was conducted at four different types, i.e., 20%, 40%, 60%, 80%, and 100%, with a constant shear rate of 0.05/s. Each set of above tests was carried out at 0 kA/m, 96 kA/m,194 kA/m, and 391 kA/m, respectively. Furthermore, before each test, the sample was pre-sheared at 1/s for 10 s first and then let stand for 30 s. All the tests were performed at 25 °C unless otherwise stated.

## 3. Result and Discussion

### 3.1. Rheology in Quasi-Static Monotonic Shear

#### 3.1.1. Quasi-Static Monotonic Shear for MR Grease with Different CI Particles Content

Figure 4 shows the shear stress–shear strain curves for MR grease with different CI particle contents at the shear rate of 0.5/s. Whether in the absence or presence of magnetic field strength, shear stress under quasi-static shear increases with the increase of CI particle content in MR grease, which is similar to the stress under dynamic shear [10,20]. Before entering into a steady plastic flow, shear stress exhibits a steady slow increase with shear strain in zero magnetic field strength. When the external magnetic field strength was applied, a sudden increase in shear stress at the initial stage appears, which is attributed to the flow resistance caused by the magneto-induced structure of MR grease. Another noticeable characteristic of stress versus strain curves is that shear stress of some samples demonstrates a sudden increase at first, followed by a decrease and then trend gently (yielding behavior) at specified magnetic field strength, i.e., both MRG-50 and MRG-70 at the magnetic field strength of 96 kA/m, MRG-70 at the magnetic field strength of 194 kA/m. However, at the magnetic field strength of 391 kA/m, there was no yielding behavior observed. The reason for this could be related to the near solid-state of MR grease, which is harder and more brittle due to the stronger chain or cluster structures formed at higher magnetic field strength.

#### 3.1.2. Quasi-Static Monotonic Shear for MR Grease under Different Shear Rate

The relationship between shear stress and shear strain under different shear rates and external magnetic field for MRG-70 are shown in Figure 5. At the same magnetic field strength condition, the shear strain dependence of shear stress under different shear rates is similar. For example, at the zero-field condition, with the increase of shear strain, shear stress of MRG-70 under different shear rate gradually trends gently after steady growth. When the external magnetic field strength was applied, shear stress of MRG-70 under different shear rates rapidly reached a plateau after a sharp increase. Another common characteristic presented in Figure 5 is that shear stress increases with shear rate. From Figure 5a, in the absence of magnetic field strength, shear stress of MRG-70 is less affected by the shear rate at a small shear strain. With the enlargement of shear strain, the effect of shear rate on shear stress becomes large. After the shear strain increased to 20%, the shear rate has a significant impact on shear stress, i.e., shear stress of MRG-70 increased from 0.075 kPa to 0.15 kPa, as high as 100%, with the shear rate increased from 0.005/s to 0.1/s. From Figure 5b–d, the shear stress was only moderately affected by shear rate under the external magnetic field strength. Especially at the high magnetic field strength of 391 kA/m, the shear rate had almost no impact on shear stress. It can be attributed to the influence of the damping to the overall stress of the material since at high magnetic field the material behaves more like a solid.

To investigate the influence of the shear rate on the yield stress of MR grease under different magnetic field strength. Maximum yield stress versus shear rate curves for MR grease with various percentages of CI particles are illustrated in Figure 6. The maximum yield stress increased with shear rate, and this phenomenon is more obvious at the zero-field condition. The shear rate-induced maximum yield stress increase of MR grease is summarized in Table 2. From Table 2, in the absence of magnetic field strength, the shear rate-induced maximum yield stress increase of MRG-30, MRG-50, and MRG-70 were 241%, 95.2%, and 110%, respectively. When the magnetic field strength of 391 kA/m was applied, the shear rate-induced maximum yield stress increase for different MR grease sharply decreased. Especially for MRG-70, shear rate-induced maximum stress increase is only 2.5%. The above feature demonstrates that the shear rate dependence of maximum yield stress is greatly weakened by the magnetic field, and this weakening is further enhanced as the CI particles content of MR grease increased.

Figure 7 shows the dependence of the maximum yield stress on magnetic field strength for different MR grease at the fixed shear rate. The maximum yield stress shows a rapid increase at first and then slowly saturates. Furthermore, maximum yield stress was greatly affected by CI particles content of MR grease, i.e., for MRG-70, the rate of increase for the maximum yield stress is as high as 6367% in the magnetic field strength range from 0 kA/m to 740 kA/m, while the yield stress growth rate of MRG-30 was 1455%. The highest maximum yield stress of MRG-30, MRG-50, and MRG-70 at the magnetic field strength of 740 kA/m were 1.5 kPa, 4.7 kPa, and 8.7 kPa, respectively. The above phenomenon demonstrates that the stronger magneto-induced chain or cluster structures are formed at high CI particles content and magnetic field strength.

#### 3.1.3. Quasi-Static Monotonic Shear for MR Grease under Different Temperature

The shear stress of MRG-70 as a function of shear strain under different temperature and magnetic field strength at the fixed shear rate of 0.05 s^−1^ is shown in Figure 8. Shear stress shows a decreased trend with the increase of temperature when the external magnetic field strength was not applied. With magnetic field strength, e.g., 391 kA/m, shear stress, especially at high shear strain, was almost unaffected by temperature, which is similar to the temperature-dependent properties of grease under the shear mode of steady and oscillatory [10]. The reason for this phenomenon is that the temperature-sensitive grease matrix dominates the quasi-static properties at the zero-field condition. With the presence of the magnetic field, the CI particles structure becomes the main character in terms of shear performance, and the matrix only has a secondary influence. The energy required to break the chain structure is a few hundred times higher than that needed for the matrix itself and temperature in this sense has little impact on the chain structure.

### 3.2. Rheology in Quasi-Static Cyclic Shear

#### 3.2.1. Quasi-Static Cyclic Shear for MR Grease with Different CI Particles Content

As known, the hysteretic loop area of the shear stress–strain curve indicates the capacity of energy-absorbing [25], which can be used to measure the damping performance of MR grease. The loop area can be calculated with integral formula as follows:(1)$$D=\oint_{L_{\mathrm{ss}-\mathrm{st}}}\tau d\gamma ,$$
where *D* represents the hysteretic loop area of the shear stress–strain curve. *τ* and *γ* represent shear stress and shear strain, respectively. *L*_ss-st_ is the shear stress versus shear strain curve under different excitation condition.

Figure 9 shows the cyclic shear stress–strain curves for MR grease with different CI particle contents under the various magnetic field strengths at the constant shear rate of 0.05/s. In the absence of the magnetic field, the cyclic shear stress–strain curves of MR grease exhibit an ellipse-like shape. When an external magnetic field is applied, especially at 391 kA/m, a parallelogram-like shape was demonstrated by the MR grease with different CI particles content, which indicates that the microstructure of MR grease changes with the application of a magnetic field. The loop area of cyclic shear stress–strain curves shown in Figure 9 are calculated and present in Table 3. The influence of the particle content on the damping properties is also calculated, respectively. From Table 3, the loop area of MR grease increases with CI particles content, and the higher the magnetic field strength, the larger the particles content-induced effect, i.e., 38.5%, 146%, 235%, and 428% for 0 kA/m, 96 kA/m, 194 kA/m, and 391 kA/m, respectively. In addition, the loop area of MR grease also increases with magnetic field strength, the magneto-induced effect of MRG-70 at 0.05/s can be up to 5703% as the applied strength increase from 0 to 391 kA/m. It is shown that the damping performance of MR grease can be enhanced by increasing CI particles content and magnetic field strength.

To further investigate the shape-changing regularity of the cyclic shear stress–strain curves of three types of MR grease under various magnetic field strength at the shear rate of 0.05/s, the shear stress–strain curves of three types of MR grease shown in Figure 9 are separately plotted in different graphs, as seen in Figure 10. It can be seen from Figure 10 that, in the absence of a magnetic field, the cyclic shear stress–strain curves of three type MR grease all display an ellipse-like shape. When the magnetic field strength of 96 kA/m was applied, the shape of the cyclic curves for MRG-30 immediately change from ellipse-like to parallelogram-like, but a polygon-like shape is demonstrated by the MR grease with higher CI particles content, i.e., MRG-50 and MRG-70. With the further increase of magnetic field strength, the shape of the cyclic curves for MRG-30 gradually develop into rectangle-like, and MRG-50 and MRG-70 also change to parallelogram-like shape. The microstructure of MR grease changes can be used to explain the reason for the different shape shown in the above, i.e., at a relatively low magnetic field strength, the magneto-induced microstructure of MR grease with lower CI particles are more stable than that of MR grease with higher CI particles content. With the further increase of magnetic field strength, the magneto-induced microstructures of MR grease with different CI particles content all tend to stabilize gradually.

Another interesting phenomenon we can find from the cyclic shear stress–strain curves marked by the blue dotted line in Figure 10 is that, when the magnetic field strength was applied to MR grease, shear stress of MRG-50 and MRG-70 change sharply during the transition from loading (shear strain varies from 0 to 100% or 0 to −100%) to unloading (shear strain varies from 100% to 0 or −100% to 0), but this phenomenon is not apparent for MRG-30. This phenomenon has never been reported in MR fluid or MR elastomer. The reason for this may be sourced from the unique viscoelastic properties of grease matrix of MR grease. Table 4 shows the stress change rate of MRG-50 and MRG-70 during the transition from loading to unloading under different magnetic field. From Table 3, shear stress change rate of MRG-50 is almost constant along with the magnetic field strength, while that of MRG-70 shows a decreasing trend. In addition, shear stress change rate of MRG-70 is always higher than that of MRG-50.

#### 3.2.2. Quasi-Static Cyclic Shear for MR Grease with Different Shear Rate

Figure 11 shows the shear rate dependent cyclic stress–strain curves of MRG-70 under different magnetic field strength. When the magnetic field strength was not applied, an ellipse-like shape was also present by the cyclic stress curves for MRG-70 under different shear rate. In the presence of the magnetic field strength of 96 kA/m, we can see from the cyclic curves marked by blue dotted line that, the cyclic shear stress–strain curves of MRG-70 under different shear rate show a different changing regularity at the beginning unloading stages (shear strain change from 100% to 82%), i.e., shear stress at the shear rate of 0.1/s remains almost constant along shear strain, but for the shear rate of 0.033/s, 0.02/s, and 0.005/s, shear stress gradually exhibits a linear increase with the shear strain. With the magnetic field strength increased to 194 kA/m, the different changing regularity is shown above only occurs in the shear strain change from 100% to 87%. When the magnetic field strength of 391 kA/m was applied, the different changing regularity disappeared. This phenomenon can be explained as, at the beginning stage of unloading, the magneto-induced chain structures experience complex break and reformation changes again under the external shear and magnetic field strength. When the external magnetic field strength is low, and the shear rate is high, the magneto-induced chain structures of MR grease cannot be simultaneously recovered while being destroyed due to the resistance of the grease matrix. However, with the magnetic field strength as high as 391 kA/m, the high magnetic field strength allows the chain structures of MR grease recover in a very short time, which make cyclic stress–strain curves nearly independent of shear rate.

The Loop area of MRG-70 at different magnetic field strength and shear rate shown in Table 5 was calculated from the cyclic stress–strain curves in Figure 11. From Table 5, the loop area of MRG-70 shows an increasing trend as the shear rate increase from 0.005/s to 0.1/s, and this increasing trend decreases rapidly with the increase of external magnetic field strength, i.e., when the external magnetic field strength is zero, the shear rate-induced effect of MRG 70 is as high as 73%. As the external magnetic field strength increases to 391 kA/m, the shear rate-induced effect of MRG-70 drops rapidly to 6.6%. In addition, the magneto-induced effect of MRG-70 decreases with the increase of shear rate in the range from 0.005/s to 0.1/s, 8530%, 6458%, 5979%, and 5061% for 0.005/s, 0.02/s, 0.033/s, and 0.1/s, respectively. The results shown above indicate that the rate-dependent damping performance of MR grease can be weakened by increasing the external magnetic field strength.

#### 3.2.3. Quasi-Static Cyclic Shear for MR Grease with Different Shear Strain

Figure 12 shows the cyclic shear stress–strain curves under different shear strain and magnetic field strength at the constant shear rate of 0.05/s. From Figure 12, the cyclic shear–strain curve shape of MRG-70 under different shear strain is elliptical-like in the zero magnetic field condition but polygonal-like in the presence of the magnetic field, which is same as that of MR grease at different CI particles content or shear rate. In addition, when the magnetic field strength is 96 kA/m, with the increase of shear strain, the cyclic loop of MRG-70 increases in lateral direction and decreases in vertical direction, after the shear strain increase to 60%, the cyclic loop of MRG-70 still increases in lateral direction, but remain constant in vertical direction. At the magnetic field strength of 194 kA/m, a similar phenomenon can also be seen when the shear strain is more significant than 20%. However, with the magnetic field strength further increase to 391 kA/m, all the cyclic loop of MRG-70 at different shear strain remain constant in the vertical direction. The above phenomenon shows that the interval of the plastic flow of MRG-70 can be enhanced by the magnetic field strength. The reason for this is that the stronger chain or cluster structures are formed at higher magnetic field strength, which makes MRG-70 harder and more brittle.

The loop area, shear strain-induced effect, and magnetic field-induced effect of stress–strain curves shown in Figure 12 are calculated and present in Table 6. From Table 6, the loop area of MRG-70 increase with the shear strain in the range from 20% to 100%, and it is interesting that the shear strain-induced effect demonstrates a rapid decrease first and then increases with the external magnetic field, 1167%, 267%, 541%, and 546% for 0 kA/m, 96 kA/m, 194 kA/m, and 391 kA/m, respectively. Moreover, in the range of shear strain from 20% to 100%, the magnetic field-induced effect decreases rapidly and followed by a constant level, i.e., 10,600%, 6556%, 5610%, 5397%, and 5360% for 20%, 40%, 60%, 80%, and 100%, respectively. It is shown that the shear strain-dependent damping performance of MR grease varies greatly with or without the applied magnetic field, which should be taken into account for the structural dimension design of MR damper.

#### 3.2.4. Quasi-Static Cyclic Shear for MR Grease under Multiple Cyclic Loading

To investigate the time-dependent performance of MR grease under quasi-static cyclic shear, a multiple cyclic loading was conducted on MRG-70 under different magnetic field strength. Figure 13 shows the cyclic shear stress–strain curves for MRG-70 under multiple cyclic loading at different magnetic field strength. In the test, the shear strain continuously and repeatedly progressed five cycles with the constant shear rate of 0.05 s^−1^ and the shear–strain amplitude of 100%. In the absence of the external magnetic field, the shear stress–strain curves tend to stable at the fourth cycle. But when the magnetic field strength of 391 kA/m was applied, the shear stress–strain curves are well repeated from the second cycle, indicating its time independence. The reason attributed to this is that a stable chain or cluster microstructure is formed under the application of the external magnetic field.

## 4. Conclusions

In this paper, MR grease with various weight percentages of CI particles was prepared by employing lithium-based grease as a continuous phase. The dependences of the magneto-induced rheological properties for MR grease on CI particles content, shear rate, and lager deformation under quasi-static monotonic/cyclic shear condition were tested and discussed. In the quasi-static monotonic shear test, it was found that, the maximum yield stress has a significant correlation with CI particles content, i.e., under the condition of shear rate of 0.05/s, the increase rate of the maximum yield stress of MRG-70 was as high as 6367% in the magnetic field strength range from 0 kA/m to 740 kA/m, while the maximum yield stress increase rate of MRG-30 was 1455%. In addition, the shear rate dependence of maximum yield stress was greatly weakened by the magnetic field, and this weakening was further enhanced as the CI particles content of MR grease increased. In the quasi-static cyclic shear test for MR grease with different CI particles content, the experiment showed that, in the absence of magnetic field, the cyclic shear stress–strain curves of three type MR grease all displayed an ellipse-like shape. However, with the increase of magnetic field strength, the shape of cyclic curves of MRG-30 gradually developed into rectangle-like, and MRG-50 and MRG-70 gradually changed to parallelogram-like shape. Moreover, shear stress of MRG-50 and MRG-70 changes sharply during the transition from loading to unloading, but this phenomenon is not evident for MRG-30. In the quasi-static cyclic shear test with different shear rate, the results showed that, at the different external magnetic field strength, the cyclic shear stress–strain curves of MRG-70 under different shear rate showed a different changing regularity at the beginning of the unloading stages, and the rate-dependent damping performance of MRG-70 decrease with the increase of the external magnetic field strength. In the quasi-static cyclic shear test with different shear strain, the results indicated that the shear strain-dependent damping performance of MR grease varied drastically with or without an applied magnetic field, which should be considered for the structural dimension design of MR dampers. It is finally remarked that a model investigation to characterize the properties of MR grease under quasi-static condition will be undertaken as the main work of the next phase. Magnetic field strength, frequency, strain amplitude, and temperature will be considered respectively.

## Figures and Tables

**Figure 1 materials-12-02431-f001:**
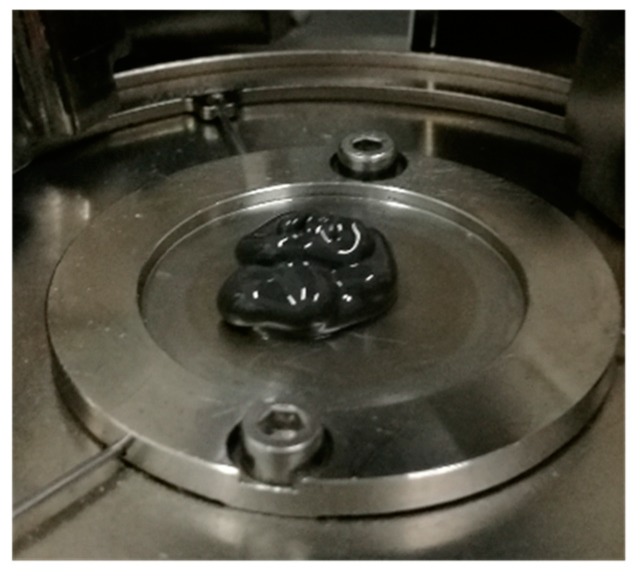
Magnetorheological (MR) grease with 70% weight percentage of CI particles.

**Figure 2 materials-12-02431-f002:**
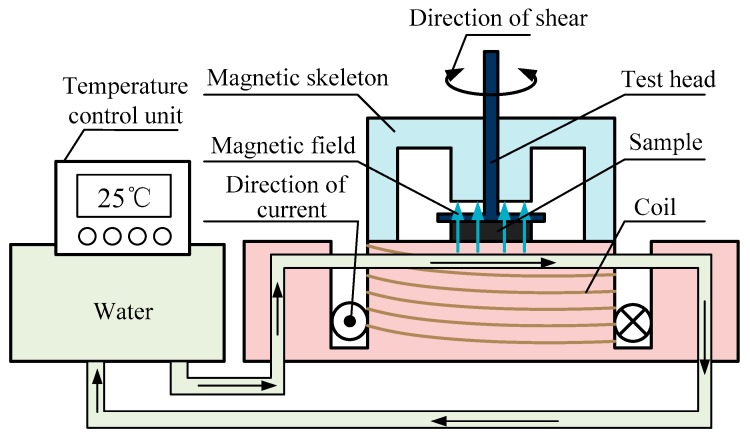
The sketch map of MCR 302 rheometer.

**Figure 3 materials-12-02431-f003:**
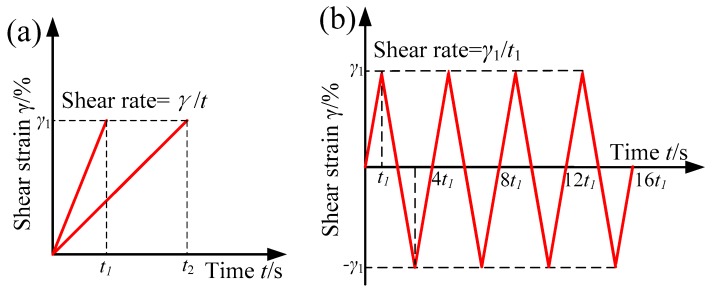
The test input of the quasi-static monotonic shear signal (**a**) and cyclic shear signal (**b**).

**Figure 4 materials-12-02431-f004:**
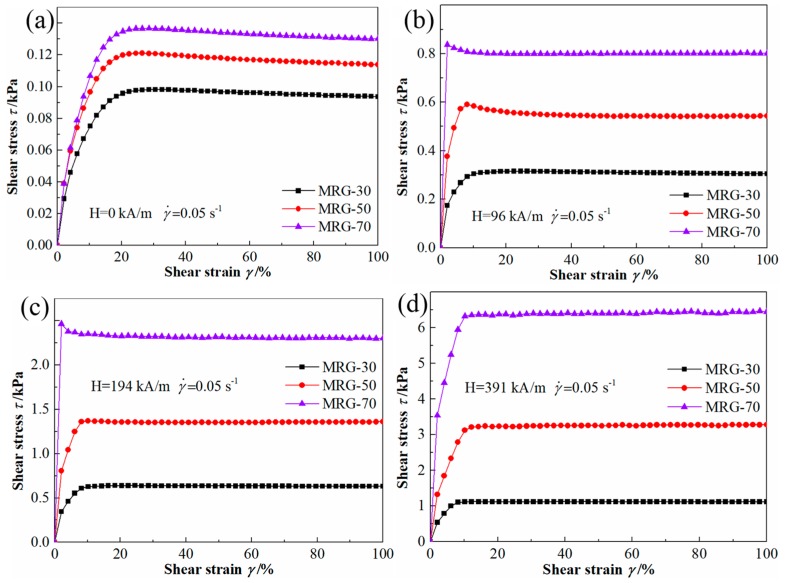
Shear stress as a function of shear strain for different MR grease at a shear rate of 0.05/s. (**a**) H = 0 kA/m; (**b**) H = 96 kA/m; (**c**) H = 194 kA/m; (**d**) H = 391 kA/m.

**Figure 5 materials-12-02431-f005:**
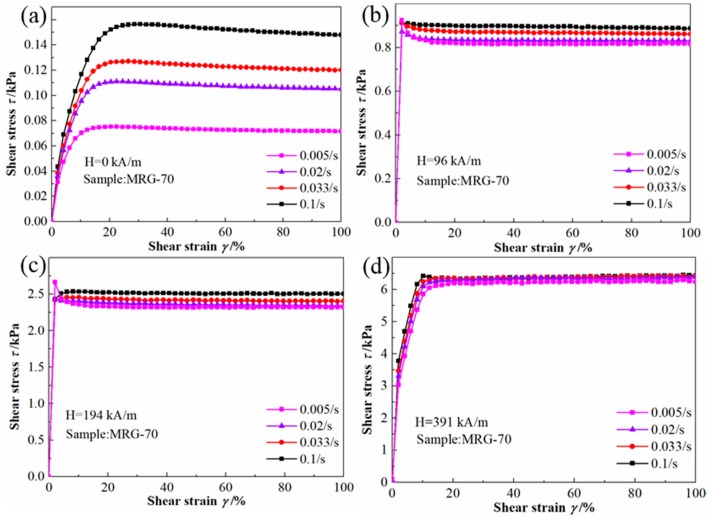
Shear stress versus shear strain for MRG-70 under different shear rate. (**a**) H = 0 kA/m; (**b**) H = 96 kA/m; (**c**) H = 194 kA/m; (**d**) H = 391 kA/m.

**Figure 6 materials-12-02431-f006:**
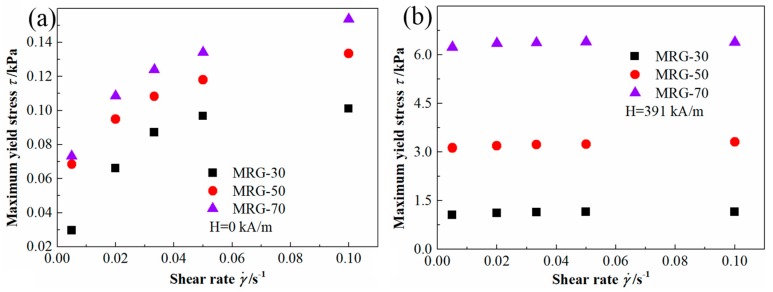
Maximum yield stress versus shear rate for different MR grease under the magnetic field strength of (**a**) H = 0 kA/m and (**b**) H = 391 kA/m.

**Figure 7 materials-12-02431-f007:**
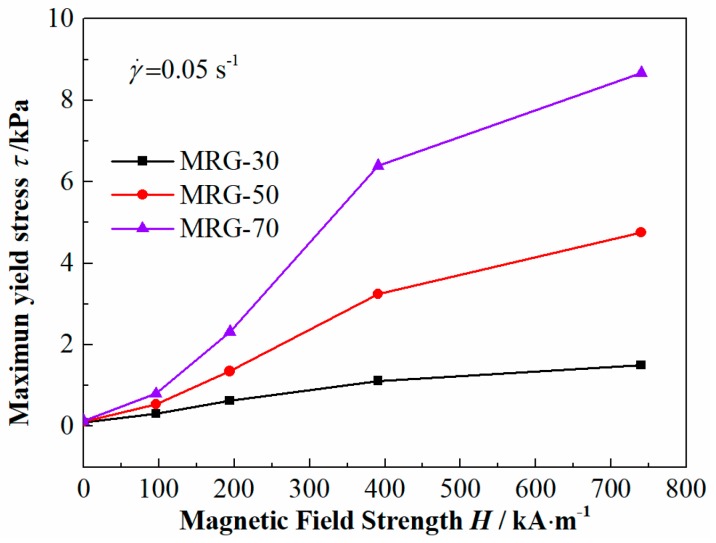
Maximum yield stress as a function of magnetic field strength for MR grease with various percentages of CI particles at the shear rate of 0.05/s.

**Figure 8 materials-12-02431-f008:**
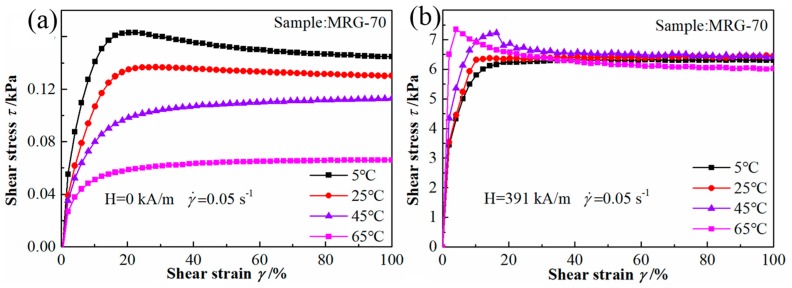
Shear stress–shear strain plots of MRG-70 under different temperature at the magnetic field strength of (**a**) H = 0 kA/m and (**b**) H = 391 kA/m.

**Figure 9 materials-12-02431-f009:**
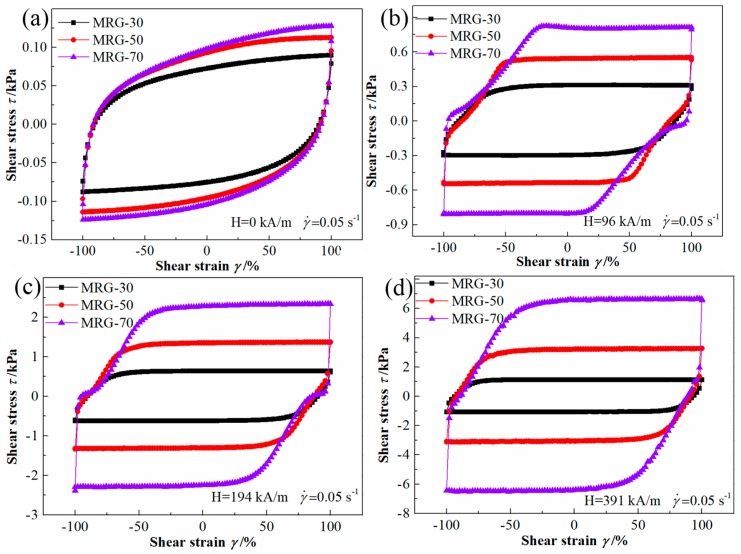
The cyclic shear stress–strain curves for different MR grease at the shear rate of 0.05/s. (**a**) H = 0 kA/m; (**b**) H = 96 kA/m; (**c**) H = 194 kA/m; (**d**) H = 391 kA/m.

**Figure 10 materials-12-02431-f010:**
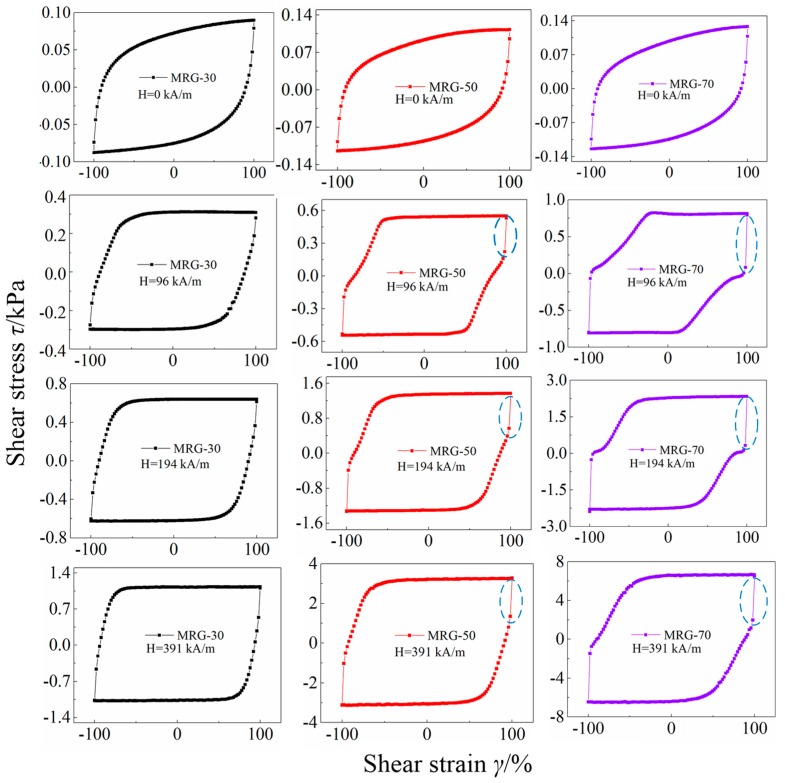
The cyclic shear stress–strain curves of different MR grease under various magnetic field strength. At the shear rate of 0.05/s.

**Figure 11 materials-12-02431-f011:**
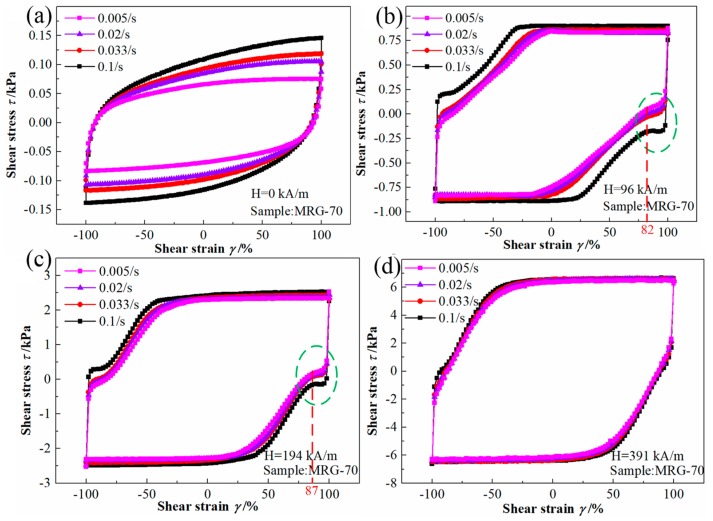
The cyclic shear stress–strain curves for MRG-70 under different shear rate. (**a**) H = 0 kA/m; (**b**) H = 96 kA/m; (**c**) H = 194 kA/m; (**d**) H = 391 kA/m.

**Figure 12 materials-12-02431-f012:**
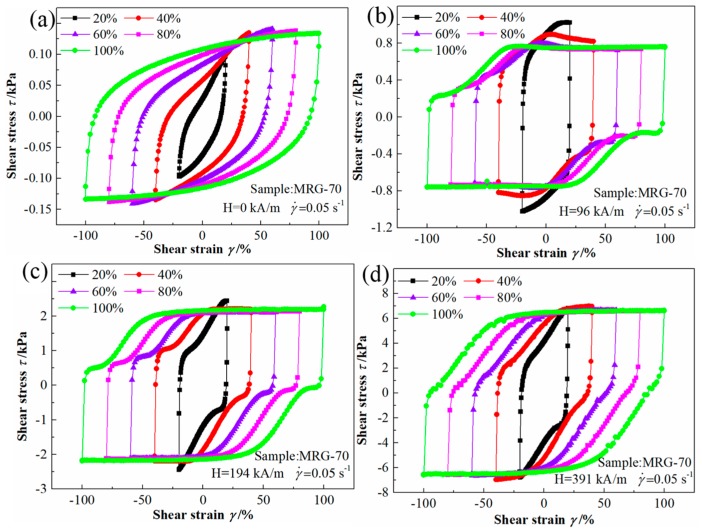
The cyclic shear stress–strain curves for MRG-70 under different shear strain. (**a**) H = 0 kA/m; (**b**) H = 96 kA/m; (**c**) H = 194 kA/m; (**d**) H = 391 kA/m.

**Figure 13 materials-12-02431-f013:**
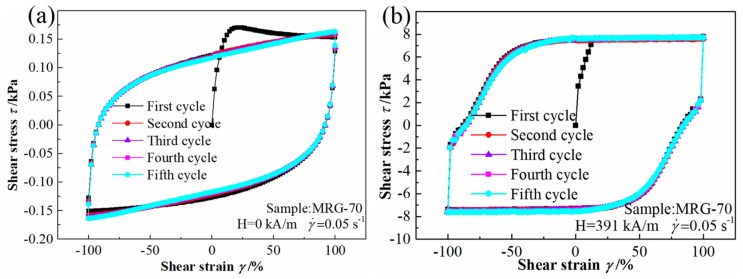
The cyclic shear stress–strain curves for MRG-70 under multiple cyclic loading. (**a**) H = 0 kA/m; (**b**) H = 391 kA/m.

**Table 1 materials-12-02431-t001:** The composition of lithium-based magnetorheological (MR) grease (wt %).

Samples	CI Particles	Lithium-based Grease
MRG-30	30	70
MRG-50	50	50
MRG-70	70	30

**Table 2 materials-12-02431-t002:** Maximum yield stress growth value and growth rate under different MR grease and magnetic field strength in the shear rate range from 0.005/s to 0.1/s.

Samples	Magnetic Field Strength	Shear Stress *τ* (Pa)		Shear Stress Increase (*τ*_0.1/s_ − *τ*_0.005/s_)/*τ*_0.005/s_ × 100%
		Shear Rate		
		0.005/s	0.1/s	
MRG-30	0 kA/m	29.6	101	241%
	391 kA/m	1060	1150.6	8.6%
MRG-50	0 kA/m	68.4	133.5	95.2%
	391 kA/m	3128.1	3306.3	5.7%
MRG-70	0 kA/m	73.2	153.5	110%
	391 kA/m	6232.6	6389.5	2.5%

**Table 3 materials-12-02431-t003:** Loop area at the shear rate of 0.05/s for different MR grease and magnetic field strength.

Magnetic Field Strength	Loop Area *D* (KJ/m^3^)			Particles Content-Induced Effect (*D*_MRG-70_ − *D*_MRG-30_)/*D*_MRG-30_ × 100%
	Sample Type			
	MRG-30	MRG-50	MRG-70	
0 kA/m	0.26	0.33	0.36	38.5%
96 kA/m	1	1.75	2.46	146%
194 kA/m	2.17	4.49	7.3	236%
391 kA/m	3.96	10.91	20.89	428%
Magneto-induced effect (*D*_391kA/m_ − *D*_0kA/m_)/*D*_0kA/m_ × 100%	1423%	3206%	5703%	-

**Table 4 materials-12-02431-t004:** Stress change rate of MRG-50 and MRG-70 during the transition from loading to unloading under different magnetic field.

MR Grease Type	Shear Stress Change Rate		
	Magnetic Field Strength		
	96 kA/m	194 kA/m	391 kA/m
MRG-50	59.6%	58.1%	58.7%
MRG-70	89.8%	85.9%	70.3%

**Table 5 materials-12-02431-t005:** Loop area of MRG-70 at different magnetic field strength and shear rate.

Magnetic Field Strength	Loop Area *D* (KJ/m^3^)				Shear Rate-Induced Effect (*D*_0.1/s_ − *D*_0.005/s_)/*D*_0.005/s_ × 100%
	Shear Rate				
	0.005/s	0.02/s	0.033/s	0.1/s	
0 kA/m	0.23	0.31	0.34	0.41	78.3%
96 kA/m	2.29	2.39	2.5	2.85	24.5%
194 kA/m	7.01	7.25	7.45	7.96	13.6%
391 kA/m	19.85	20.33	20.67	21.16	6.6%
Magneto-induced effect (*D*_391 kA/m_ − *D*_0 kA/m_)/*D*_0 kA/m_ × 100%	8530%	6458%	5979%	5061%	-

**Table 6 materials-12-02431-t006:** Loop area of MRG-70 under different magnetic field strength and shear strain.

Magnetic Field Strength	Loop Area *D* (KJ/m^3^)					Shear Strain-Induced effect (*D*_100%_ − *D*_20%_)/*D*_20%_ × 100%
	Shear Strain Amplitude					
	20%	40%	60%	80%	100%	
0 kA/m	0.03	0.11	0.2	0.29	0.38	1167%
96 kA/m	0.67	1.15	1.51	1.94	2.46	267%
194 kA/m	1.11	2.59	3.97	5.48	7.12	541%
391 kA/m	3.21	7.32	11.42	15.94	20.75	546%
Magneto-induced effect (*D*_391 kA/m_ − *D*_0 kA/m_)/*D*_0 kA/m_ × 100%	10,600%	6556%	5610%	5397%	5360%	-

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
