# Peer review of "Quasi-Static Rheological Properties of Lithium-Based Magnetorheological Grease under Large Deformation"

_materials, 2019, doi:10.3390/ma12152431_

Round 1

Reviewer 1 Report

Overall, the manuscript presented testing methodology for identifying correlation between carbonyl iron concentration and chain strength in MR grease. This subject and MR grease in general is of particular interest to the reviewer, having worked primarily with MR Fluid and had hands on experience with the shortcomings of the fluid form (e.g., leaks).

The reviewer's first comment is to have the grammar and spelling errors cleaned up throughout the paper. Often the reviewer found interpretation of text and reading between the lines were required to understand material.

The next comment applies to the Figure 9 and Figure 10. The reviewer is curious to know if the sharp change in strain indicated by dotted blue lines is indicative of a data acquisition/processing artifact. 

In general, the paper is well constructed and the results support the conclusion. The conclusion would be made stronger if future recommendations or planned future work was added. Surely, this study won't be the last as the intro implies the intent is to eventually use MR grease for engineering applications.

Reviewer 2 Report

The manuscript investigates a lithium-based magnetorheological grease (MRG), analyzing the quasi-static rheological properties.

In the reviewer's opinion, the paper is well written and technically sound.

Some minor comments are here reported:

- It is suggested to add  in section 2.1 a picture of the MR grease sample, used in the tests;

- Since the performance of the MRG could be affected by the operation time, at the end of Section 3 ("Result and discussion"), it is suggested to add a new subsection in which the authors could investigate the performance of the MRG, when the cyclic shear signal is applied for a long time  (for example ten minutes or more), without the temperature control.
